# Text-guided 3D Human Generation from 2D Collections

**Tsu-Jui Fu[1], Wenhan Xiong[2], Yixin Nie[2]**
**Jingyu Liu[2], Barlas Oğuz[2], William Yang Wang[1]**

[1]UC Santa Barbara  [2]Meta

{tsu-juifu, william}@cs.ucsb.edu  {xwhan, ynie, jingyuliu, barlaso}@meta.com

## Abstract

3D human modeling has been widely used for engaging interaction in gaming, film, and animation. The customization of these characters is crucial for creativity and scalability, which highlights the importance of controllability. In this work, we introduce Text-guided 3D Human Generation (T3H), where a model is to generate a 3D human, guided by the fashion description. There are two goals: 1) the 3D human should render articulately, and 2) its outfit is controlled by the given text. To address this T3H task, we propose Compositional Cross-modal Human (CCH). CCH adopts cross-modal attention to fuse compositional human rendering with the extracted fashion semantics. Each human body part perceives relevant textual guidance as its visual patterns. We incorporate the human prior and semantic discrimination to enhance 3D geometry transformation and fine-grained consistency, enabling this to learn from 2D collections for data efficiency. We conduct evaluations on DeepFashion and SHHQ with diverse fashion attributes covering the shape, fabric, and color of upper and lower clothing. Extensive experiments demonstrate that CCH achieves superior results for T3H with high efficiency.

## 1 Introduction

Our world is inherently three-dimensional, where this nature highlights the importance of 3D applications in various fields, including architecture, product design, and scientific simulation. The capability of 3D content generation helps bridge the gap between physical and virtual domains, providing an engaging interaction within digital media. Furthermore, realistic 3D humans have vast practical value, especially in gaming, film, and animation. Despite enhancing the user experience, the customization of the character is crucial for creativity and scalability. Language is the most direct way of communication. If a system follows the description and establishes

---

Project website: https://text-3dh.github.io

*The man is sporting a short-sleeved cotton t-shirt and pure-colored medium shorts*

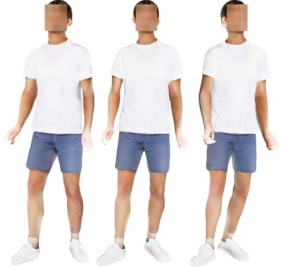

*The lady is wearing a graphic-patterned tank top, paired with three-point denim pants*

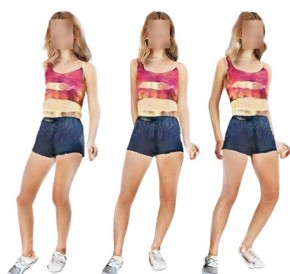

*The woman is wearing a floral-patterned long-sleeved sweater and long denim pants*

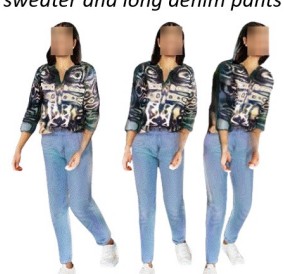

*She is dressed in a solid-colored sleeveless shirt with long cotton trousers*

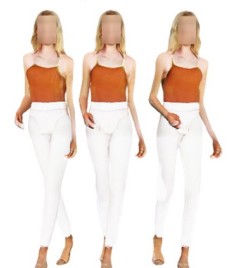

Figure 1: Text-guided 3D Human Generation (T3H).

the 3D human model, it will significantly improve controllability and meet the considerable demand.

We thus introduce Text-guided 3D Human Generation (T3H) to generate a 3D human with the customized outfit, guided via the fashion description. Previous works (Kolotouros et al., 2019; Gao et al., 2022) depend on multi-view videos to learn 3D human modeling, but these data are difficult to obtain and are not language controllable. Text-to-3D (Jain et al., 2022; Poole et al., 2023) has shown attractive 3D generation results through the success of neural rendering (Mildenhall et al., 2020). However, these methods apply iterative inference optimization by external guidance, which is inefficient for usage.

To tackle these above issues, we propose Compositional Cross-modal Human (CCH) to learn T3H from 2D collections. CCH divides the human body into different parts and employs individual volume rendering, inspired by EVA3D (Hong et al., 2023).

We extract the fashion semantics from the description and adopt cross-modal attention to fuse body volumes with textual features, where each part can learn to perceive its correlated fashion patterns. To support various angles of view, CCH leverages the human prior (Bogo et al., 2016) to guide the geometry transformation for concrete human architecture. Then these compositional volumes can jointly render a 3D human with the desired fashion efficiently. The semantic discrimination further considers compositional distinguishment over each human part, which improves the fine-grained alignment with its description through adversarial training.

We perform experiments on DeepFashion (Liu et al., 2016; Jiang et al., 2022) and SHHQ (Fu et al., 2022a), which contain human images with diverse fashion descriptions. The patterns include various types of shapes (*sleeveless*, *medium short*, *long*, etc.), fabrics (*denim*, *cotton*, *furry*, etc.), and colors (*floral*, *graphic*, *pure color*, etc.) for the upper and lower clothing. To study the performance of T3H, we conduct a thorough evaluation from both visual and semantic aspects. We treat overall realism, geometry measure, and pose correctness to assess the quality of generated 3D humans. For the alignment with the assigned fashion, we apply text-visual relevance from CLIP and fine-grained accuracy by a trained fashion classifier.

The experiments indicate that language is necessary to make 3D human generation controllable. Our proposed CCH adopts cross-modal attention to fuse compositional neural rendering with textual fashion as 3D humans, and semantic discrimination further helps fine-grained consistency. In summary, our contributions are three-fold:

- We introduce T3H to control 3D human generation via fashion description, learning from collections of 2D images.
- We propose CCH to extract fashion semantics in the text and fuse with 3D rendering in one shot, leading to an effective and efficient T3H.
- Extensive experiments show that CCH exhibits sharp 3D humans with clear textual-related fashion patterns. We advocate that T3H can become a new field of vision-and-language research.

## 2 Related Work

**Text-guided Visual Generation.** Using human-understandable language to guide visual generation can enhance controllability and benefit creative visual design. Previous works built upon adversarial training (Goodfellow et al., 2015; Reed et al., 2016) to produce images (Xu et al., 2018; El-Nouby et al., 2019; Fu et al., 2020, 2022c) or videos (Marwah et al., 2017; Li et al., 2018b; Fu et al., 2022b) conditioned on given descriptions. With sequential modeling from Transformer, vector quantization (Esser et al., 2021) can generate high-quality visual content as discrete tokens (Ramesh et al., 2021; Ding et al., 2021; Fu et al., 2023). The denoising diffusion framework (Ho et al., 2020; Ramesh et al., 2022; Saharia et al., 2022; Feng et al., 2023) gains much attention as its diversity and scalability via large-scale text-visual pre-training. Beyond images and videos, 3D content creation is more challenging due to the increasing complexity of the depth dimension and spatial consistency. In this paper, we consider text-guided 3D human generation (T3H), which has vast applications in animated characters and virtual assistants.

**3D Generation.** Different representations have been explored for 3D shapes, such as mesh (Gao et al., 2019; Nash et al., 2020; Henderson et al., 2020), voxel grid (Tatarchenko et al., 2017; Li et al., 2017), point cloud (Li et al., 2018a; Yang et al., 2019; Luo et al., 2021), and implicit field (Chen and Zhang, 2019; Park et al., 2019; Zheng et al., 2022). Neural Radiance Field (NeRF) (Mildenhall et al., 2020; Barron et al., 2022; Muller et al., 2022) has shown remarkable results in novel view synthesis (Schwarz et al., 2021; Chan et al., 2021; Skorokhodov et al., 2023) and 3D reconstruction (Yariv et al., 2021; Zhang et al., 2021). With the differentiable neural rendering, NeRF can be guided by various objectives. Text-to-3D draws appreciable attraction these days, which adopts external text-visual alignments (Jain et al., 2022; Khalid et al., 2022; Michel et al., 2022; Wang et al., 2022a; Hong et al., 2022) and pre-trained text-to-image (Wang et al., 2022b; Nam et al., 2022; Poole et al., 2023; Metzer et al., 2023; Tang et al., 2023; Seo et al., 2023). However, existing methods take numerous iterations to optimize a NeRF model, which is time-consuming for practical usage. Our CCH learns to extract fashion semantics with NeRF rendering and incorporates the human prior for a concrete human body, achieving effective and efficient T3H.

**3D Human Representation.** To reconstruct a 3D human, early works (Collet et al., 2015; Guo et al., 2019; Su et al., 2020) count on off-the-shelf tools to predict the camera depth. As mitigating the costly hardware requirement, they estimate a 3D human

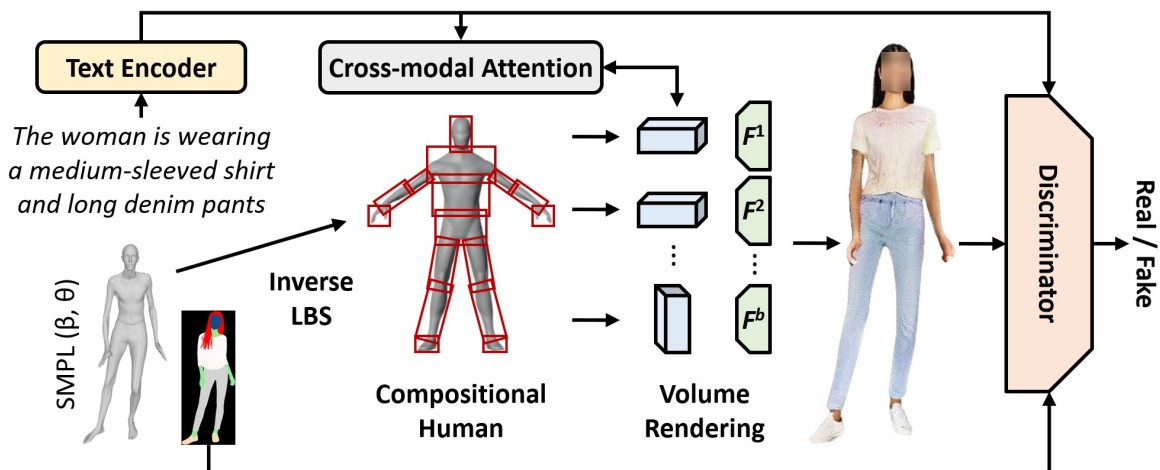

Figure 2: Compositional Cross-modal Human (CCH). CCH extracts fashion semantics from the description and adopts cross-modal attention in compositional body volumes for controllable 3D human rendering. The human prior (SMPL) provides robust geometry transformation, enabling CCH to learn from 2D collections for data efficiency. The semantic discrimination further helps find-grained consistency through adversarial training.

texture (Xu and Loy, 2021; Gomes et al., 2022) via the UV mapping (Shysheya et al., 2019; Yoon et al., 2021). With the promising success of NeRF, recent works (Peng et al., 2021b,a; Su et al., 2021) adopt volume rendering for 3D humans from multi-view videos (Weng et al., 2022; Chen et al., 2022). Since the data are difficult to collect, the 3D-aware generation (Chan et al., 2022; Gu et al., 2022; Noguchi et al., 2022) learns 3D modeling from the collection of human images (Yang et al., 2022; Hong et al., 2023). In place of arbitrary outputs, we introduce the first controllable 3D human generation that also learns from a 2D collection, and the presented fashion patterns should align with the description.

## 3 Text-guided 3D Human Generation

### 3.1 Task Definition

We present text-guided 3D human generation (T3H) to create 3D humans via fashion descriptions. For data efficiency, a 2D collection $\mathcal{D}=\{\mathcal{V},\mathcal{T}\}$ is provided, where $\mathcal{V}$ is the human image, and $\mathcal{T}$ is its fashion description. Our goal is to learn the neural rendering that maps $\mathcal{T}$ into an articulate 3D human with the fashion patterns of $\mathcal{V}$.

### 3.2 Background

Neural Radiance Field (**NeRF**) (Mildenhall et al., 2020) defines implicit 3D as $\{c,\sigma\}=F(x,d)$. The query point $x$ in the viewing direction $d$ holds the emitted radiance $c$ and the volume density $\sigma$. To get the RGB value $C(r)$ of certain rays $r(t)$, volume rendering is calculated along a ray $r$ from the near

bound $t_{\mathrm{n}}$ to the far bound $t_{\mathrm{f}}$:

$$T(t) = \exp(-\int_{t_{\mathrm{n}}}^{t} \sigma(r(s))ds),$$

$$C(r) = \int_{t_{\mathrm{n}}}^{t_{\mathrm{f}}} T(t)\sigma(r(t))c(r(t),d)dt, \quad (1)$$

where $T(t)$ stands for their accumulated transmittance. StyleSDF (Or-El et al., 2022) then replaces $\sigma$ with single distance field (SDF) $d(x)$ for a better surface, where $\sigma(x) = \alpha^{-1}\mathrm{sigmoid}(\frac{-d(x)}{\alpha})$ and $\alpha$ is a learnable scalar that controls the tightness of the density around the surface boundary.

**SMPL** (Bogo et al., 2016) defines the human body as $\{\beta,\theta\}$, where $\beta \in \mathbb{R}^{10}$ and $\theta \in \mathbb{R}^{3\times23}$ control its shape and pose. We consider Linear Blend Skinning (LBS) as the transformation from the canonical into the observation space for the point $x$ to $\sum_{k=1}^{K} h_k H_k(\theta, J)x$, where $h_k \in \mathbb{R}$ is the scalar of the blend weight and $H_k \in \mathbb{R}^{4\times4}$ is the transformation matrix of the $k$th joint. Inverse LBS transforms the observation back to the canonical space as a similar equation but with an inverted $H$.

### 3.3 Compositional Cross-modal Human

Following EVA3D (Hong et al., 2023), we split the human body into 16 parts. As shown in Fig. 2, each body part holds its own bounding box $\{o_{\mathrm{min}}^{b}, o_{\mathrm{max}}^{b}\}$. To leverage the human prior for a target pose $\theta$, we transform these pre-defined bounding boxes with SMPL's transformation matrices $H_k$. Ray $r(t)$ is sampled for each pixel on the canvas. For a ray that intersects bounding boxes, we pick up its near and far bounds ($t_{\mathrm{n}}$ and $t_{\mathrm{f}}$) and sample $N$ points as follows: $t_i \sim \mathcal{U}\left[t_{\mathrm{n}} + \frac{i-1}{N}(t_{\mathrm{f}} - t_{\mathrm{n}}), t_{\mathrm{n}} + \frac{i}{N}(t_{\mathrm{f}} - t_{\mathrm{n}})\right]$.

We then transform these sampled points back to the canonical space with inverse LBS. For shape generalization, we consider not only pose transformation but also blend shapes ($B^P(\theta)$ and $B^S(\beta)$) (Zheng et al., 2021). $\mathbb{N}$ contains $K$ nearest vertices $v$ of the target SMPL mesh for the sample point ray $r(t_i)$:

$$g_k = \frac{1}{||r(t_i) - v_k||},$$
$$M_k = \left(\sum_{k=1}^{K} g_k H_k\right) \begin{bmatrix} I & B_k^P + B_k^S \\ 0 & I \end{bmatrix},$$
$$\begin{bmatrix} x_i \\ 1 \end{bmatrix} = \sum_{v_k \in \mathbb{N}} \frac{g_k}{\sum_{v_k \in \mathbb{N}} g_k} (M_k)^{-1} \begin{bmatrix} r(t_i) \\ 1 \end{bmatrix}, \quad (2)$$

where $g_k \in \mathbb{R}$ is the inverse weight of the vertex $v_k$ and $M_k \in \mathbb{R}^{4 \times 4}$ is the transformation matrix. The $x_i$ can be used for further volume rendering.

**Cross-modal Attention.** During rendering, if the canonical point $x_i$ with the viewing direction $d_i$ is inside the $b$th bounding box, it will be treated as:

$$\hat{x}_i^b = \frac{2x_i - (o_{max}^b + o_{min}^b)}{o_{max}^b - o_{min}^b},$$
$$f_i^b = \text{Linear}(\hat{x}_i^b, d_i), \quad (3)$$

where a linear mapping is applied to acquire preliminary features $f^b \in \mathbb{R}^{16 \times 8 \times 128}$. To exhibit the desired fashion in the final rendering, we extract the word features by the text encoder as $\{w_l \in \mathbb{R}^{512}\}$ from $\mathcal{T}$. We then fuse the textual features with $f_i^k$ via cross-modal attention:

$$p_l = \frac{\exp(f_i^b W^b w_l^\mathrm{T})}{\sum_{\iota=1}^{L} \exp(f_i^b W^b w_\iota^\mathrm{T})},$$
$$\text{CA}(f_i^b \mid \{w\}) = \sum_{l=1}^{L} p_l w_l, \quad (4)$$

where $L$ is the length of $\mathcal{T}$ and $W^b$ is the learnable matrix. In this way, each point can learn to perceive relevant textual guidance for the $b$th human body part and depict corresponding fashion patterns.

Each body part has its individual volume rendering $F^b$, which consists of stacked multilayer perceptrons (MLPs) with the SIREN activation (Sitzmann et al., 2020). Since the point $x_i$ may fall into multiple boxes $\mathbb{B}_i$, we follow EVA3D to apply the mixture function (Lombardi et al., 2021):

$$\{c_i^b, \sigma_i^b\} = F^b(\text{CA}(x_i^b, d_i \mid \{w\})),$$
$$u_b = \exp(-m(\hat{x}_i^b(x)^n + \hat{x}_i^b(y)^n + \hat{x}_i^b(z)^n)),$$
$$\{c_i, \sigma_i\} = \frac{1}{\sum_{b \in \mathbb{B}} u_b} \sum_{b \in \mathbb{B}} u_b \{c_i^b, \sigma_i^b\}, \quad (5)$$

---

**Algorithm 1** Compositional Cross-modal Human

1: $\mathcal{D}$: 2D collection of human images / fashion descriptions
2:
3: $G, D$: the generator / discriminator model for T3H
4: **while** TRAIN_CCH **do**
5:      $\mathcal{V}, \mathcal{T} \leftarrow$ sampled human / description from $\mathcal{D}$
6:      $\{\beta, \theta\} \leftarrow$ estimated SMPL parameters of $\mathcal{V}$
7:      $\{x_i\} \leftarrow$ canonical points via inverse LBS   $\triangleright$ Eq. 2
8:      $f_i^b \leftarrow$ rendering features inside the $b$th box  $\triangleright$ Eq. 3
9:      $\{w_l\} \leftarrow$ extracted textual features of $\mathcal{T}$
10:     CA $\leftarrow$ fusion via cross-modal attention   $\triangleright$ Eq. 4
11:     $\{c_i, \sigma_i\} \leftarrow$ mixture radiance / density   $\triangleright$ Eq. 5
12:     $R \leftarrow$ final rendering human   $\triangleright$ Eq. 1
13:
14:     $\mathcal{S} \leftarrow$ segmentation map of $\mathcal{V}$
15:     $Q \leftarrow$ fashion map between $\mathcal{S}$ and $\mathcal{T}$   $\triangleright$ Eq. 6
16:     $\mathcal{L}_{adv} \leftarrow$ adversarial loss from $D$   $\triangleright$ Eq. 8
17:     $\mathcal{L}_{off}, \mathcal{L}_{eik} \leftarrow$ offset and derivation loss  $\triangleright$ Eq. 9
18:     $\mathcal{L}_{all} \leftarrow$ overall training loss   $\triangleright$ Eq. 10
19:     Update $G$ by minimizing $\mathcal{L}_{all}$
20:     Update $D$ by maximizing $\mathcal{L}_{all}$
21: **end while**

---

where $m$ and $n$ are hyperparameters. With $\{c_i, \sigma_i\}$, we adopt Eq. 1 to render the RGB value of ray $r(t)$. Through all sampled rays $r$, we then have our final human rendering $R$, where the overall process can be simplified as $R = G(\beta, \theta \mid \mathcal{T})$. To summarize, CCH leverages the human prior and adopts inverse LBS to acquire the canonical space for the target pose. The human body is divided into 16 parts, and each of them fuses its correlated fashion semantics via cross-modal attention. Finally, compositional bodies jointly render the target 3D human.

**Semantic Discrimination.** With the SMPL prior, our CCH contains robust geometry transformation for humans and can learn from 2D images without actual 3D guidance. For a ground-truth $\{\mathcal{V}, \mathcal{T}\}$, we parse the 2D human image as the segmentation $\mathcal{S}$ (MMHuman3D, 2021), which provides the reliable body architecture. To obtain its fashion map $Q$, we apply cross-modal attention between $\mathcal{S}$ and $\mathcal{T}$:

$$\{e_{i,j}\} = \text{Conv}(\mathcal{S}),$$
$$Q_{i,j} = \sum_{l=1}^{L} \frac{\exp(e_{i,j} W w_l^\mathrm{T})}{\sum_{\iota=1}^{L} \exp(e_{i,j} W w_\iota^\mathrm{T})} w_l, \quad (6)$$

where $e$ is the same dimension as $f$, $W$ is the learnable attention matrix, and $Q$ perceives which human body part should showcase what fashion patterns. We concatenate the rendered human $R$ (or the ground-truth $\mathcal{V}$) with $Q$ and feed them into our discriminator $D$ to perform binary classification:

$$D(R \mid \mathcal{T}) = \text{BC}([\text{Conv}(R), Q]). \quad (7)$$

In consequence, $D$ can provide alignments of both

| Method | DeepFashion | | | | | SHHQ | | | | |
| --- | --- | --- | --- | --- | --- | --- | --- | --- | --- | --- |
| | FID↓ | Depth↓ | PCK↑ | CLIP-S↑ | FA↑ | FID↓ | Depth↓ | PCK↑ | CLIP-S↑ | FA↑ |
| Latent-NeRF | 69.654 | 0.0298 | 74.211 | 22.500 | 65.883 | 72.256 | 0.0381 | 73.401 | 22.210 | 67.427 |
| TEXTure | 37.058 | 0.0165 | 86.354 | _23.385_ | _67.508_ | 48.618 | 0.0216 | 85.502 | _24.456_ | _68.233_ |
| CLIP-O | _25.488_ | _0.0133_ | _87.892_ | 21.887 | 61.964 | _34.212_ | **0.0164** | _87.312_ | 21.401 | 66.808 |
| CCH | **21.136** | **0.0121** | **88.355** | **25.023** | **72.038** | **32.858** | _0.0165_ | **87.624** | **27.855** | **76.194** |

Table 1: Overall results of pose-guided T3H.

the human pose and fashion semantics, which improves the fine-grained consistency of our CCH.

**Learning of CCH.** We include the non-saturating loss with R1 regularization (Mescheder et al., 2018) for adversarial learning over the ground-truth $\{\mathcal{V}\}$:

$$U(u) = -\log(1 + \exp(-u)),$$
$$\mathcal{L}_{\text{adv}} = U(G(\beta, \theta \mid \mathcal{T}) \mid \mathcal{T}) \quad (8)$$
$$+ U(-D(\mathcal{V} \mid \mathcal{T})) + \lambda|\nabla D(\mathcal{V} \mid \mathcal{T})|^2.$$

Following EVA3D, we also append the minimum offset loss $\mathcal{L}_{\text{off}}$ to maintain a plausible human shape as the template mesh. $\mathcal{L}_{\text{eik}}$ penalizes the derivation of delta SDFs to zero and makes the estimated SDF physically valid (Gropp et al., 2020):

$$\mathcal{L}_{\text{off}} = ||\Delta d(x)||_2^2,$$
$$\mathcal{L}_{\text{eik}} = ||\nabla(\Delta d(x))||_2^2. \quad (9)$$

The learning process of our CCH is also illustrated as Algo. 1, where the overall optimization can be:

$$\mathcal{L}_{\text{all}} = \mathcal{L}_{\text{adv}} + 1.5 \cdot \mathcal{L}_{\text{off}} + 0.5 \cdot \mathcal{L}_{\text{eik}}, \quad (10)$$
$$\min_G \max_D \mathcal{L}_{\text{all}}.$$

# 4 Experiemnts

## 4.1 Experimental Setup

**Datasets.** We coduct experiments on DeepFashion (Jiang et al., 2022) and SHHQ (Fu et al., 2022a) for T3H. DeepFashion contains 12K human images with upper and lower clothing descriptions. Since there are no annotations in SHHQ, we first fine-tune GIT (Wang et al., 2022c) on DeepFashion and then label for 40K text-human pairs. We follow Open-Pose (Cao et al., 2019) and SMPLify-X (Pavlakos et al., 2019) to estimate the human keypoints and its SMPL parameters. The resolution is resized into 512x256 in our experiments. Note that all faces in datasets are blurred prior to training, and the model is not able to generate human faces.

**Evaluation Metrics.** We apply metrics from both visual and semantic prospects. Following EVA3D, we adopt Frechet Inception Distance (**FID**) (Heusel et al., 2017) and **Depth** (Ranftl et al., 2020) to cal-

culate visual and geometry similarity, compared to the ground-truth image. We treat Percentage of Correct Keypoints (**PCK@0.5**) (Andriluka et al., 2014) as the correctness of the generated pose. To investigate the textual relevance of T3H results, we follow CLIP Score (**CLIP-S**) (Hessel et al., 2021) for the text-visual similarity. We fine-tune CLIP (Radford et al., 2021) on DeepFashion for a more accurate alignment in this specific domain. To have the fine-grained evaluation, we train a fashion classifier on DeepFashion labels[1] and assess Fashion Accuracy (**FA**) of the generated human.

**Baselines.** As a new task, we consider the following methods as the compared baselines.

- Latent-NeRF (Metzer et al., 2023) brings NeRF to the latent space and guides its generation by the given object and a text-to-image prior.
- TEXTure (Richardson et al., 2023) paints a 3D object from different viewpoints via leveraging the pre-trained depth-to-image diffusion model.
- CLIP-O is inspired by AvatarCLIP (Hong et al., 2022), which customizes a human avatar from the description with CLIP text-visual alignment. We apply the guided loss to optimize a pre-trained EVA3D (Hong et al., 2023) for faster inference.
- Texformer (Xu and Loy, 2021) estimates the human texture from an image. Text2Human (Jiang et al., 2022) predicts the target human image, and we treat Texformer to further build its 3D model.

For a fair comparison, all baselines are re-trained on face-blurred datasets and cannot produce identifiable human faces.

**Implementation Detail.** We divide a human body into 16 parts and deploy individual StyleSDF (Or-El et al., 2022) for each volume rendering, and two following MLPs then estimate SDF and RGB values. We adopt the same discriminator as StyleSDF over fashion maps to distinguish between fake rendered humans and real images. We sample $N$=28 points for each ray and set $(m, n)$ to (4, 8) for mix-

---

[1] There are six targets for FA, including the shape, fabric, and color of the upper and lower clothing. The fashion classifier has +95% accuracy, which provides a precise evaluation.

| | DeepFashion | | |
|---|---|---|---|
| Method | FID↓ | CLIP-S↑ | FA↑ |
| Texformer | 45.844 | 20.546 | 66.679 |
| CLIP-O | 25.579 | 20.112 | 61.298 |
| CCH | **21.355** | **24.920** | **70.771** |

Table 2: Overall results of pose-free T3H.

| Ablation Settings | | | DeepFashion | | |
|---|---|---|---|---|---|
| Text | CA | SD | FID↓ | CLIP-S↑ | FA↑ |
| ✗ | ✗ | ✗ | 25.671 | 9.632 | 36.634 |
| ✓ | ✗ | ✗ | 24.624 | 21.079 | 69.173 |
| ✓ | ✓ | ✗ | 21.966 | 24.103 | 80.028 |
| ✓ | ✓ | ✓ | **21.275** | **25.211** | **80.776** |

Table 3: Ablation study (256x128) of Cross-modal Attention (CA) and Semantic Discrimination (SD).

| | | DeepFashion | | |
|---|---|---|---|---|
| Train | FT. | R1 | R5 | R10 |
| OpenAI-400M | ✗ | 4.2 | 20.4 | 28.0 |
| LAION-2B | ✗ | 13.4 | 33.4 | 46.4 |
| LAION-2B | ✓ | **45.0** | **83.0** | **93.8** |

Table 4: Text-to-Fashion retrieval (sample 500 pairs) by CLIP with different fine-tunings (FT.).

| | DeepFashion | |
|---|---|---|
| Method | Quality | Relevance |
| Latent-NeRF | 1.82 | 2.37 |
| TEXTure | 2.38 | 2.51 |
| CLIP-O | **2.93** | 2.20 |
| CCH | 2.87 | **2.92** |

Table 5: Human evaluation for T3H with aspects of 3D quality and fashion relevance.

ture rendering. The text encoder is initialized from CLIP and subsequently trained with CCH. We treat Adam (Kingma and Ba, 2015) with a batch size of 1, where the learning rates are 2e-5 for $G$ and 2e-4 for $D$. We apply visual augmentations by randomly panning, scaling, and rotating within small ranges. All trainings are done using PyTorch (Paszke et al., 2017) on 8 NVIDIA A100 GPUs for 1M iterations.

## 4.2 Quantitative Results

Table 1 shows the pose-guided T3H results on Deep-Fashion and SHHQ, where we feed the estimated human mesh as the input object into Latent-NeRF and TEXTure. Although Latent-NeRF can portray body shapes in multiple angles from its latent NeRF space, the rendering is clearly counterfeit (higher FID and Depth). For TEXTure, the human architecture is constructed well by the given mesh (higher PCK). However, the estimated texture is still spatially inconsistent and contains inevitable artifacts (still higher FID). From the semantic aspect, Latent-NeRF and TEXTure borrow those trained diffusion models and depict the assigned appearance in the description (higher CLIP-S than CLIP-O). CLIP-O relies on EVA3D to produce feasible 3D humans (lower FID). While the external CLIP loss attempts to guide the fashion, the global alignment is insufficient to demonstrate detailed patterns (lower FA). Without those above drawbacks, our CCH learns to extract fashion semantics along with the compositional human generation, leading to comprehensive superiority across all metrics.

A similar trend can be found on SHHQ. Latent-NeRF and TEXTure exhibit related fashion patterns but are hard to present realistic humans (higher FID and Depth). CLIP-O produces a sharp human body

with the correct pose, but not the assigned fashion (lower CLIP-S and FA) by the inexplicit alignment from CLIP. Table 2 presents the pose-free results. With the guided 2D image, Texformer contains the assigned clothing in the text (higher FA than CLIP-O). But the 3D reconstruction is unable to handle spatial rendering, resulting in low-quality humans (higher FID). With cross-modal attention and semantic discrimination, CCH exceeds baselines in both visual and textual relevance, making concrete human rendering with the corresponding fashion.

## 4.3 Ablation Study

We study each component effect of CCH in Table 3. Without the guided description, the model lacks the target fashion and results in a poor FA. This further highlights the importance of textual guidance for controllable human generation. When applying the traditional training (Reed et al., 2016), conditional GAN is insufficient to extract fashion semantics for effective T3H (not good enough CLIP-S). On the other hand, our cross-modal attention constructs a better fusion between fashion patterns and volume rendering, facilitating a significant improvement in depicting the desired human appearance. Moreover, semantic discrimination benefits fine-grained alignment and leads to comprehensive advancement.

**Fine-tune CLIP-S as Evaluator.** CLIP has shown promising text-visual alignment, which can calculate feature similarity between the generated human and the given text as CLIP-S (Hessel et al., 2021). Since our T3H is in a specific fashion domain, we consider the larger-scaled trained checkpoint from OpenCLIP (Ilharco et al., 2021) and fine-tune it as

*The woman is wearing a short-sleeved t-shirt, paired with three-point denim pants*

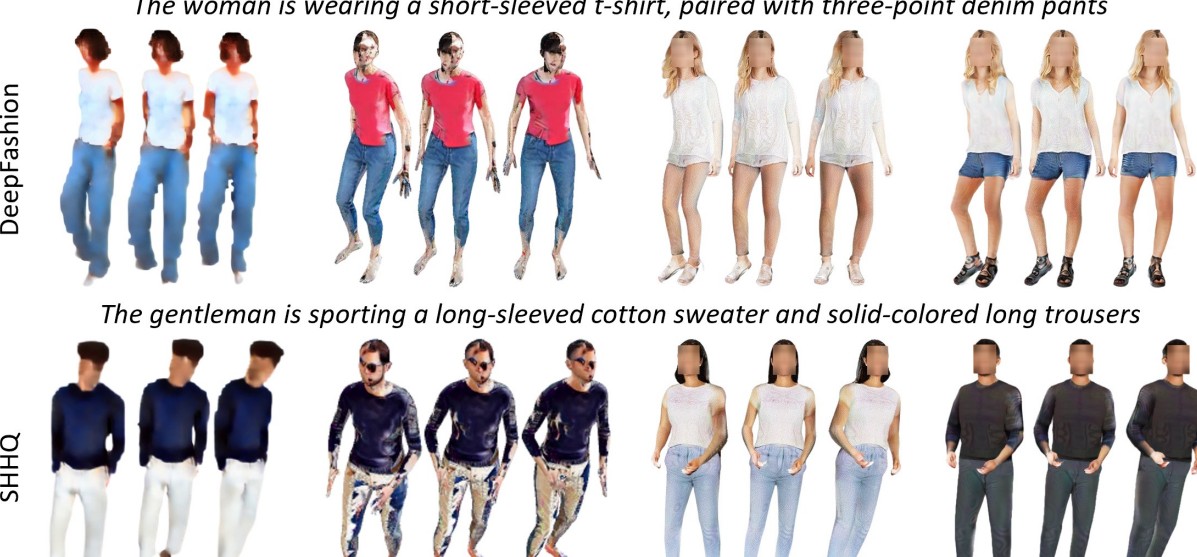

Figure 3: Qualitative comparison of pose-guided T3H.

| Method | Time (sec) | GPU (MB) |
|---|---|---|
| Latent-NeRF | 755.7 | 11250 |
| TEXTure | 103.7 | 12530 |
| CLIP-O | 181.6 | 15988 |
| CCH | **0.372** | **6258** |

Table 6: Time and GPU cost to perform T3H.

a more precise evaluator. Table 4 presents text-to-fashion retrieval results, where a higher recall leads to a better alignment. Whether the original CLIP or OpenCLIP, both result in poor performance and is insufficient for our evaluation. By perceiving Deep-Fashion, fine-tuning helps bring reliable alignment and is treated as the final evaluator.

**Human Evaluation.** Apart from automatic metrics, we conduct the human evaluation with aspects of 3D quality and fashion relevance. We randomly sample 75 T3H results and consider MTurk[2] to rank over baselines and our CCH. To avoid the potential ranking bias, we hire 3 MTurkers for each example. Table 5 shows the mean ranking score (from 1 to 4, the higher is the better). CLIP-O and CCH are built upon EVA3D, which provides an articulate human body for superior 3D quality. Even if Latent-NeRF and TEXTure take pre-trained diffusion models to acquire visual guidance, CCH exhibits more corresponding fashion via cross-modal fusion. This performance trend is similar to our evaluation, which supports the usage of CLIP-S and FA as metrics.

**Inference Efficiency.** In addition to T3H quality,

---

[2]Amazon MTurk: https://www.mturk.com

*She is dressed in a sleeveless tank top with long denim pants*

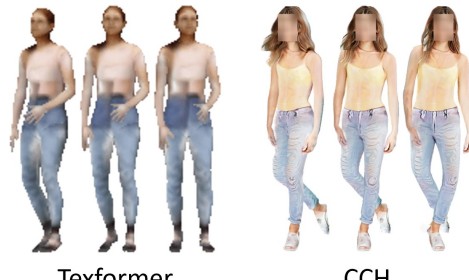

Figure 4: Qualitative comparison of pose-free T3H.

our CCH also contains a higher efficiency. Table 6 shows the inference time and GPU cost on a single NVIDIA TITAN RTX. All baselines take more than 100 seconds since they require multiple iterations to optimize the 3D model from an external alignment. In contrast, we extract fashion semantics and carry out T3H in one shot. Without updating the model, we save the most GPU memory. In summary, CCH surpasses baselines on both quality and efficiency, leading to an effective and practical T3H.

### 4.4 Qualitative Results.

We demonstrate the qualitative comparison of pose-guided T3H in Fig. 3. Although Latent-NeRF can portray the 3D human based on the given mesh, it only presents inauthentic rendering. TEXTure generates concrete humans, but there are still obvious cracks and inconsistent textures from different angles of view. Moreover, both of them fail to capture "*three-point*", where the rendered lower clothing is

*This man is wearing a striped medium sleeve and long cotton trousers*

*She is dressed in a long-sleeved chiffon shirt with striped three-point shorts*

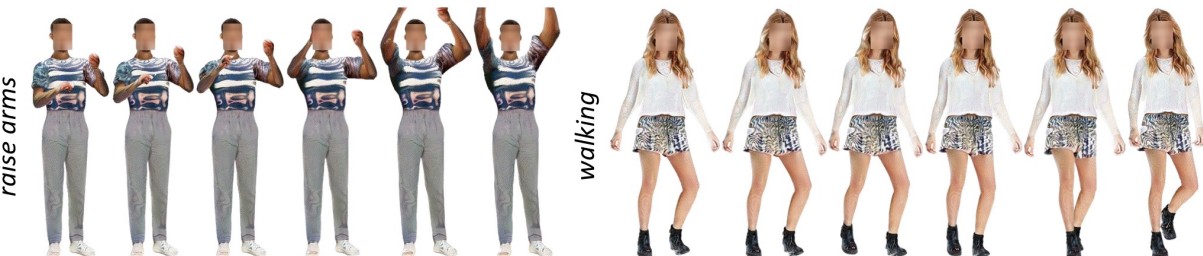

raise arms

walking

Figure 5: Qualitative examples of animatable T3H, where the motion is also controlled by the text.

*The woman is wearing a striped cotton shirt, paired with long cotton pants*

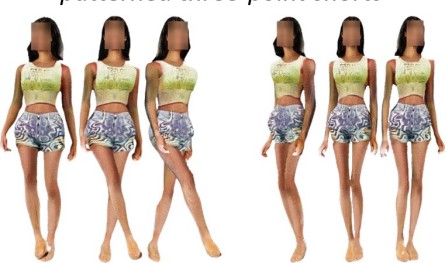

*She is sporting a graphic-patterned tank top with floral-patterned three-point shorts*

Figure 6: Qualitative examples of pose-control T3H.

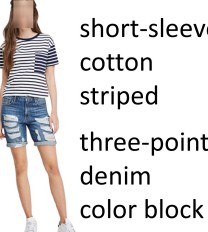 short-sleeve cotton striped

three-point denim color block

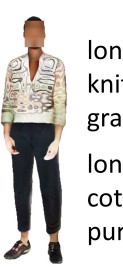 long-sleeve knitted graphic

long cotton pure color

Figure 7: Qualitative examples of our fashion classifier, which provides fine-grained labels for real/fake humans.

human body present diverse appearances; different poses then guide the character to express rich body language. This flexibility in controlling appearance and pose allows for better practical customization.

**Animatable T3H.** In addition to static poses, CCH can benefit from dynamic motions to achieve animatable T3H. Fig. 5 adopts MotionDiffuse (Zhang et al., 2022) to create the assigned action also from the text and apply it to our produced 3D models. In this way, we prompt them to "*raise arms*" or "*walk*" for favorable dynamic scenarios.

## 5 Conclusion

We present text-guided 3D human generation (T3H) to create a 3D human by a fashion description. To learn this from 2D collections, we introduce Compositional Cross-modal Human (CCH). With cross-modal attention, CCH fuses compositional human rendering and textual semantics to build a concrete body architecture with the corresponding fashion. Experiments across various fashion attributes show that CCH effectively carries out T3H with high efficiency. We believe T3H helps advance a new field toward vision-and-language research.

**Ethics Discussion and Limitation.** Our work enhances the controllability of 3D human generation. To prevent identity leakage, we blur out faces prior to training and avoid risks similar to DeepFake (Korshunov and Marcel, 2018). Because we depend on SMPL parameters, an inaccurate estimation causes

incorrectly depicted as long pants. Because CLIP provides an overall but inexplicit alignment to the description, CLIP-O is limited and exhibits vague "*denim*" or "*long-sleeved*". This observation further indicates the flaw of CLIP in detailed fashion patterns, even if it has been fine-tuned on the target dataset. In contrast, our CCH adopts cross-modal attention with NeRF, contributing to high-quality T3H with fine-grained fashion controllability. Fig. 4 shows the pose-free results. Texformer relies on a 2D image to estimate its 3D texture. Despite containing the assigned fashion, it is still restricted by the capability of 3D reconstruction, resulting in a low-resolution rendering. By learning text-to-3D directly, CCH can produce textual-related humans from random poses with clear visual patterns.

**Pose-control T3H.** Since our CCH is generating 3D humans from given SMPL parameters, as illustrated in Fig. 6, we can further control T3H with a specific pose. Different fashion descriptions make a

a distribution shift and quality degradation. For the datasets, they reveal narrow viewing angles, which results in visible artifacts of 3D consistency.

**Acknowledgements.** We appreciate the anonymous reviewers for constructive feedback. The research presented in this work was funded by Meta AI. The views expressed are those of the authors and do not reflect the official policy or position of the funding agency.

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
