# OpenReview forum: "Text-guided 3D Human Generation from 2D Collections"
_EMNLP/2023/Conference — EMNLP 2023 Findings_

### Official Review · Reviewer_o3tu · 2023-08-04

**Soundness:** 5

**Excitement:**

4: Strong: This paper deepens the understanding of some phenomenon or lowers the barriers to an existing research direction.

**Paper Topic And Main Contributions:**

This paper addresses the method of generating 3D human body models from fashion descriptions. Several challenges in previous studies, known as T3H, had problems with unclear rendering, the use of hard-to-obtain multi-view videos for training, lack of language control, and time-consuming iterative inference optimization.

The main contributions towards solving these issues are as follows:

- Proposal of a method for generating 3D human models controlled by textual descriptions using 2D collections for training.
- Enhancing the efficiency of 3D human model generation by extracting fashion semantics from the text and fusing them with 3D rendering using cross-modal attention.


**Reasons To Accept:**

The strengths of this paper lie in its ability to achieve faster execution compared to conventional methods, generate high-quality 3D human models, and facilitate easier data collection for learning.

The inference time for 3D human generation, which used to take over 100 seconds in conventional methods, has been reduced to less than one second. Additionally, this reduction in computation also saves GPU memory, making it accessible to a broader audience.
The quality of the generated 3D humans is quantitatively evaluated using various metrics, demonstrating significantly better performance compared to conventional methods. This trend is consistent even in human evaluation.
The fact that the paper is capable of developing T3H, which is a new field of vision-and-language research, is the NLP community’s benefit.


**Reasons To Reject:**

This is just a minor point, but I'm not sure if the automatic evaluation criteria used in the paper are really helpful, as they do not directly measure the quality of generated models. As the paper presents a human evaluation, this may not be a very big problem, but some discussion on this point (whether they are good enough or not) may be helpful.

**Reproducibility:**

3: Could reproduce the results with some difficulty. The settings of parameters are underspecified or subjectively determined; the training/evaluation data are not widely available.

**Reviewer Confidence:**

3: Pretty sure, but there's a chance I missed something. Although I have a good feel for this area in general, I did not carefully check the paper's details, e.g., the math, experimental design, or novelty.

---

> ### Author Rebuttal · Authors · 2023-08-25
>
> Since no single criterion can cover them all, we try our best to consider various metrics to evaluate each aspect of text-guided 3D human generation.
> + FID quantifies the general realism of visual patterns;
> + Depth and PCK measure the geometry similarity of 3D humans;
> + CLIP-S and FA calculate the text-visual alignment with input descriptions.
>
> We further compute Pearson correlation between human evaluation and automatic metrics (metric scores are normalized):
> |  | 3D Quality |
> | :-: | :-: |
> | FID | 76.4 |
> | Depth | 73.8 |
> | PCK | 70.3 |
>
> |  | Fashion Relevance |
> | :-: | :-: |
> | CLIP-S | 78.0 |
> | FA | 83.2 |
>
> The results demonstrate that they are correlated, which supports our usage of automatic metrics. We will add this discussion to our revision.

---

### Official Review · Reviewer_3mSQ · 2023-08-05

**Soundness:** 4

**Excitement:**

4: Strong: This paper deepens the understanding of some phenomenon or lowers the barriers to an existing research direction.

**Paper Topic And Main Contributions:**

This paper proposed a text based 3D photorealistic human generation approch, the generation quality is good and the text controlling effectiveness is good.

**Reasons To Accept:**

1, computing efficience is better than SOTA
2, generation quality is good
3, the proposed text based controlling is effective
4, it is fancy to see the proposed approch works with motion generation algorithm.

**Reasons To Reject:**

the employed compositional 3D human rendering result has none identity information, it means no dicriminative input is provided for the neural rendering network, could result in lower robustness for final result generation.

**Reproducibility:**

4: Could mostly reproduce the results, but there may be some variation because of sample variance or minor variations in their interpretation of the protocol or method.

**Reviewer Confidence:**

4: Quite sure. I tried to check the important points carefully. It's unlikely, though conceivable, that I missed something that should affect my ratings.

---

> ### Author Rebuttal · Authors · 2023-08-25
>
> Thanks for pointing this out! We train a gender classifier on our blurred human dataset (validation accuracy >99.6%). We then adopt it to evaluate our generated 3D humans, and the gender accuracy is 91.3%. This indicates that our model can still capture the gender concept only from the visual pattern, even if without actual faces. We will add this result to our revision.

---

### Official Review · Reviewer_oudz · 2023-08-06

**Soundness:** 1

**Excitement:**

1: Poor: I cannot identify the contributions of this paper, or I believe the claims are not sufficiently backed up by evidence. I would fight to have it rejected.

**Paper Topic And Main Contributions:**

The main task of the paper is fashion description-guided 3D human generation. Since 3D human modeling is widely used in gaming and animation, as it is necessary to customize characters, the paper suggested text as the most direct way to control it. The proposed algorithm converts SMPL into a canonical space through inverse LBS, renders it to generate features that fit the given fashion description through a cross-modal attention in that space, and then sends it back to the observation space using LBS.
Although this paper utilizes text, in my opinion, it does not fit the scope of EMNLP mentioned above.


**Reasons To Accept:**

- This paper presents a new task to create a 3D human based on a fashion description.
- Since no study has performed 3D human generation based on a fashion description, the authors compared the performance with similar studies, and the experiment showed better performance based on 5 metrics in 2 datasets.


**Reasons To Reject:**

- Customizing 3D characters using text can have advantages, but it is less persuasive as to why the range of text should be limited to fashion description. For fashion, most of them are guided by images, such as virtual try on [1], and the benefits of using text for guidance are not clear.
[1] Choi, S., Park, S., Lee, M., & Choo, J. (2021). Viton-hd: High-resolution virtual try-on via misalignment-aware normalization. In Proceedings of the IEEE/CVF conference on computer vision and pattern recognition (pp. 14131-14140).
- In addition, since this study generates multi-view images based on NeRF, a video is essential for evaluating the 3D creation.
- The paper uses a cross-modal attention to create text-reflected images in multi-view, but since cross-attention is already a widely-used method in diffusion models, I do not think that it bears clear novelty.
- Also, the writing quality leaves room for improvement. For example, the math notations are inaccurate, reducing the readability of the paper. It is strongly recommended to provide a more detailed network architecture figure to aid the understanding of the proposed approach.
- Finally, although this study utilizes text, I am not sure whether it falls under the scope of this society.


**Reproducibility:**

3: Could reproduce the results with some difficulty. The settings of parameters are underspecified or subjectively determined; the training/evaluation data are not widely available.

**Reviewer Confidence:**

3: Pretty sure, but there's a chance I missed something. Although I have a good feel for this area in general, I did not carefully check the paper's details, e.g., the math, experimental design, or novelty.

**Typos Grammar Style And Presentation Improvements:**

In the paper, the mathematical notations are unconventional, leading to reduced readability. For instance, both vectors and scalars are denoted using the same symbols, and the dimensions of variables are not specified, further complicating comprehension.

---

> ### Author Rebuttal · Authors · 2023-08-25
>
> **Q1: range of text**\
> **A1:** We first want to clarify that our proposed method is not limited to fashion generation. With the flexible cross-modal attention over volume rendering, the model is able to learn from diverse text-annotated human images for 3D humans through our strategy. Our T3H is motivated by Text2Human [1,2,3], which also considers the fashion scenario. As the abundant visual data and human labels of fashion datasets (our used DeepFashion and SHHQ), it can support large-scale experiments and fine-grained evaluations. On the other hand, there is also text-guided virtual try-on [4], which also presents beneficial controllability, which is aligned with the motivation in our work.
>
> [1] (SIGGRAPH’22) Text2Human: Text-Driven Controllable Human Image Generation\
> [2] (arXiv’23) Text–Conditioned Fashion Image Editing With Guided GAN Inversion\
> [3] (arXiv’23) Text-Driven Human Video Generation\
> [4] (SIGGRAPH’23) Controllable Virtual Try-on with Text and Texture
>
> **Q2: video is essential**\
> **A2:** For human evaluation, we do present the generated human GIFs for comparison. We also provide the animated qualitative examples (also in GIF) on our project website (L40).
>
> **Q3: cross-modal attention**\
> **A3:** To the best knowledge, we are the first to integrate cross-modal attention into neural volume rendering, which can generate a 3D human in one step. In contrast, the baselines (Latent-NeRF and TEXTure) leverage a pre-trained text-to-image model, which adopts cross-modal attention only for static images, and requires numerous steps to guide a NeRF model. Hence our method shows better generated results (Tables 1 and 5) as well as significantly superior efficiency (Table 6). This further demonstrates our contribution of cross-modal attention to 3D human generation.
>
> **Q4: math notations**\
> **A4:**\
> NeRF:
> + $\alpha \in \mathbb{R}$ is a learned scalar in StyleSDF that controls the tightness of the density around the surface boundary. $\alpha=0$ represents a solid, sharp, and object boundary. A larger $\alpha$ leads to a more fluffy object boundary.
>
> SMPL:
> + $\beta \in \mathbb{R}^{10}$ controls the shape (10 parameters) of a human body.
> + $\theta \in \mathbb{R}^{69}$ controls the pose (69 parameters of 23 relative joints) of a human body.
> + $H_k \in \mathbb{R}^{4 \times 4}$ is the transformation matrix for the $k$th local joint, and $h_k \in \mathbb{R}$ is a scalar of the blend weight.
>
> Inverse LBS:
> + $M_k \in \mathbb{R}^{4 \times 4}$ and $g_k \in \mathbb{R}$ are similar to $H_k$ and $h_k$ but for th inverse transformation.
>
> Cross-modal Attention:
> + $f^b \in \mathbb{R}^{\text{h} \times \text{w} \times \text{c}}$ is the preliminary visual features, where $\text{h}=16$, $\text{w}=8$, and $\text{c}=128$.
> + $w_l \in \mathbb{R}^{\text{x}}$ is the encoded word features from the CLIP text encoder, where $\text{x}=512$.
> + $W^b \in \mathbb{R}^{\text{c} \times \text{x}}$ is the learnable attention matrix for cross-modal fusion.
>
> Semantic Discrimination:
> + $e \in \mathbb{R}^{\text{h} \times \text{w} \times \text{c}}$ is the visual features of the input segmentation map $\mathcal{S}$, which has a similar dimension of $f$.
> + $W \in \mathbb{R}^{\text{c} \times \text{x}}$ is another learnable attention matrix for the Discriminator $D$.
>
> We will add the above clarification and improve Figure 2 for a detailed architecture and training pipeline in our revision.
>
> **Q5: under the scope of EMNLP**\
> **A5:** In this paper, we investigate the controllability of text for efficient 3D human generation, which embraces the cooperation between language and visual synthesis. As you comment that "no study has performed 3D human generation based on a fashion description", we believe our introduced task can advance a new field of vision-and-language research. In addition, there are many works [1,2,3,4] of text-to-image generation/manipulation presented in recent NLP conferences. Our T3H takes a forward step from 2D to 3D, which can benefit the community.
>
> [1] (ACL’23) Building Domain-Specific Text-to-Image Synthesizers with Fast Inference Speed\
> [2] (ACL’23) Resolving Ambiguities in Text-to-Image Generative Models\
> [3] (ACL’23) I Spy a Metaphor: Large Language Models and Diffusion Models Co-Create Visual Metaphors\
> [4] (EACL’23) Improving Cross-modal Alignment for Text-Guided Image Inpainting

---

### Meta-Review · Area_Chair_KdGk · 2023-09-18

**Recommendation:** 4

**Metareview:**

Three reviewers provided feedback for this paper. They found the work interesting, the model capable and they appreciated the faster performance by the proposed approach. There was one glaring complaint from a reviewer, that had to do with fit to EMNLP. First, the reviewer felt that text conditioning the 3D generation was not sufficient to be accepted at a top language conference. Second, the reviewer felt that image conditioning was more relevant than text conditioning for this task. And third, they questioned the novelty. The authors provided a detailed rebuttal to these points and I agree with the authors views here. I find this work to be of interest to the EMNLP audience, particularly as multimodal AI is becoming increasingly popular at EMNLP. I think that text conditioned avatar generation / 3D generation is going to be very popular as a research topic going forward, and so I think this paper is very timely. I also disagree that the paper does not deliver on novelty of interest to the reader. Given the reviews, rebuttal and the above details, I recommend acceptance.

---

### Decision · Program_Chairs · 2023-10-07

**Decision:**

Accept-Findings

**Comment:**

Three reviewers provided feedback for this paper. They found the work interesting, the model capable and they appreciated the faster performance by the proposed approach. There was one glaring complaint from a reviewer, that had to do with fit to EMNLP. First, the reviewer felt that text conditioning the 3D generation was not sufficient to be accepted at a top language conference. Second, the reviewer felt that image conditioning was more relevant than text conditioning for this task. And third, they questioned the novelty. The authors provided a detailed rebuttal to these points and I agree with the authors views here. I find this work to be of interest to the EMNLP audience, particularly as multimodal AI is becoming increasingly popular at EMNLP. I think that text conditioned avatar generation / 3D generation is going to be very popular as a research topic going forward, and so I think this paper is very timely. I also disagree that the paper does not deliver on novelty of interest to the reader. Given the reviews, rebuttal and the above details, I recommend acceptance.